# Concrete/Glass Construction and Demolition Waste (CDW) Synergies in Ternary Eco-Cement-Paste Mineralogy

**DOI:** 10.3390/ma15134661

**Published:** 2022-07-02

**Authors:** Raquel Vigil de la Villa Mencía, Moises Frías, Sagrario Martínez Ramírez, Lucía Fernandez Carrasco, Rosario García Giménez

**Affiliations:** 1Departamento de Geología y Geoquímica, Geomateriales Unidad Asociada CSIC-UAM, Universidad Autónoma de Madrid, 28049 Madrid, Spain; raquel.vigil@uam.es; 2Eduardo Torroja Institute for Construction Science (IETcc-CSIC), 28033 Madrid, Spain; mfris@ietcc.csic.es; 3Institute for the Structure of Matter (IEM-CSIC), 28006 Madrid, Spain; sagrario@iem.cfmac.csic.es; 4Department of Civil and Environmental Engineering, Barcelona TECH, Universitat Politécnica de Catalunya, 08034 Barcelona, Spain; lucia.fernandez@upc.edu

**Keywords:** construction and demolition waste (CDW), recycled concrete, glass, pozzolan synergies, ternary eco-cements, mineralogical phases

## Abstract

The study described sought further understanding of the synergies in a mix of CDW pozzolans, containing (calcareous and siliceous) concrete and glass waste, used to prepare ternary eco-cement paste bearing 7% of the binary blend at concrete/glass ratios of 2:1 and 1:2. The mineralogical phases in the 2-day, 28-day, and 90-day cement matrices were identified and monitored using XRF, XRD-Rietveld, SEM-EDX, FT-IR, and NMR. The findings showed that changes in the reaction kinetics in the ternary blended pastes relative to OPC pastes depended on the nature of the recycled concrete and the glass content. Adding the binary mix bearing calcareous concrete (at a ratio of 2:1) favoured ettringite, portlandite, and amorphous phase formation, whilst the blends with siliceous concrete favoured C-S-H gel formation. Monocarboaluminate was detected in the 90-day siliceous concrete and glass pastes in amounts similar to those found in the reference OPC paste.

## 1. Introduction

Cement manufacture, which accounts for 12% to 15% of all industrial energy consumed worldwide and 5% to 7% of carbon dioxide (CO_2_) emissions, is one of the priority targets for lowering energy consumption and the generation of greenhouse gases [1].

One avenue for enhancing energy and environmental efficiency, while raising cement sustainability, is to use industrial waste as a partial replacement. Worldwide waste management has come to acquire vital importance, for the steep growth in the demand for natural resources is placing enormous pressure on the environment and society. One outcome has been the coining of new phrases in cement nomenclature such as green cement, the name applied to binders with a low clinker content and smaller carbon footprint than ordinary Portland cement (OPC). All such new eco-cements bear industrial waste as partial cement replacements. In light of their importance, these new cements are being studied as a priority line of research in initiatives such as the European Commission’s circular economy strategies, the European Environment Agency’s Green Deal, the United Nations’ Agenda 2030, and the general pursuit of carbon neutrality by 2050.

Existing European legislation [2] addresses the partial replacement of cement with mineralogical additions from industrial waste, focusing in particular on those exhibiting pozzolanicity and which are consequently usable for the manufacture of binary and ternary eco-cements. Blended cements are known to be mechanically stronger and more durable than unadditioned binders [3,4,5,6].

The vast volumes of construction and demolition waste (CDW) generated globally is one of the types of industrial waste that has attracted the greatest researcher attention in recent years. Such waste is primarily applied as a recycled aggregate in eco-mortar and eco-concrete manufacture, which requires huge quantities of such materials [7,8,9,10,11]. The use of concrete waste as coarse aggregate is presently envisaged in many countries’ structural legislation and codes [12,13,14,15].

Concrete waste is crushed and sieved to different particle size distributions for commercialisation. Coarse aggregate is used for the aforementioned applications as well as for other industrial activities (roads, concretes, embankments), whilst, for want of any industrial use, the fines (< 5mm) are simply stockpiled, unprotected in management plant yards.

A number of research teams [16,17,18,19,20,21], studying the scientific–technical viability of partially replacing cement with such fines, reported no substantial change in end-product chemistry or strength relative to OPC.

Due to their moderate pozzolanicity, further benefit might be drawn from fines if blended with highly pozzolanic mineralogical additions such as silica fume, activated rice-husk ash, bagasse ash, and glass [20,21,22]. Glass is a non-crystalline, non-biodegradable ceramic that owes its high silica content (and indirectly, its properties) to non-crystalline silicates bearing variable amounts of oxides such as CaO, Na_2_O, K_2_O, and Al_2_O_3_ [23,24]. Approximately 10 Mt of today’s worldwide glass output of 130 Mt [25] end up in landfills.

Soda–lime glass, the predominant type of glass, contains 66% to 75% SiO_2_, 12% to 17% Na_2_O, and 6% to 12% CaO [26]. Its high reactive silica content informs its speedy pozzolanic reaction when added to cement, with the formation of secondary C-S-H gels in addition to those generated during anhydrous cement phase hydration [27,28].

Recent studies have attested to the substantial synergies between glass and concrete CDW fines, where the primary hydration product has been identified as C-S-H gels with sodium in their composition. Such products are characterised by a more highly polymerised structure than those generated in the absence of waste [27].

To carry that line of research further, this study explored the synergies between two types of pozzolanic CDWs in a pozzolan/cement system bearing 7% of the addition at concrete/glass ratios of 1:2 and 2:1. The respective pastes were cured up to a total of 90 days. The ternary cements were characterised and the hydrated phases monitored using X-ray fluorescence (XRF), laser granulometry, Rietveld-refined X-ray diffraction (XRD), Fourier-transform infrared spectroscopy (FT-IR), nuclear magnetic resonance spectroscopy (NMR), thermogravimetric analysis/differential thermal analysis (TG/DTA), and scanning electron microscopy/energy dispersive X-ray analysis (SEM/EDX). The respective findings will play a significant role in the microstructure and engineering performance of future eco-mortars and eco-concretes.

## 2. Materials and Methods

### 2.1. Materials

Three types of CDW were chosen to prepare binary blends for the ternary cement pastes. Two consisted in fines (<5 mm) stockpiled at management plant yards, one in the region of Madrid, the by-product of processing waste concrete with natural siliceous aggregate (HsT), and the other at a yard in the Basque Country resulting from processing waste concrete from calcareous aggregate (HcG). The third waste was glass collected from a demolition site involving a residential building.

The ternary cement pastes (labelled as shown in Table 1) were prepared with a binder in which 7% of commercial cement type CEM I 42.5R (Cementos Lemona, S.A., Lemoa, Bilbao, Spain) was replaced with the binary blend of recycled CDW pozzolans containing different portions of concrete fines and glass. The 7% replacement ratio was defined on the grounds of previous research [16,27].

The XRF-determined chemical composition of the resulting binders is listed in Table 2 and their particle size distribution parameters are D10, D50, and D90 in Table 3.

Cement matrix hydration reactivity at a constant water/binder ratio of 0.5 was determined on 1 × 1 × 6 cm^3^ prismatic specimens following Koch–Steinegger methodology. The hydration reaction was detained after 2 days, 28 days, or 90 days by soaking the specimens in acetone for 24 h, followed by vacuum drying for a further 24 h.

### 2.2. Methods

The majority oxides in the blended and control cements were quantified on pressed wafers with a Philips PW-1404 X-ray fluorescence spectrometer fitted with a Sc-Mo dual anode tube, Philips, Eindhoven, The Netherlands.

Variation in cement TG/DTA was measured in 40 mg to 50 mg samples on a Texas Instruments 600 SDT analyser (New Castle, DE) in a nitrogen atmosphere, ramping the temperature at a rate of 10 °C/min.

Cement fineness was found with dry dispersion laser diffraction particle-size analysis on a Malvern Mastersizer 3000 analyser (Pananalytical, Madrid, Spain) bearing red and blue (He-Ne and LED) light sources. The measuring range was 0.01 µm to 3500 μm.

The mineralogical phases were identified with a PANalytical X’Pert PRO diffractometer (Madrid, Spain) fitted with a copper anode, a 0.5° divergence slit, and 0.5 mm receiving slit, operating at 40 mA and 45 kV. Samples were studied over a 2θ angle range of 5° to 60°. The powder method was applied, using 5% rutile as the internal standard to quantify the crystalline phases, and the proportion of amorphous matter in the samples. Quantification was performed with Match v.3 and Rietveld analysis with Fullprof software (Crystal Impact, Dr. H. Putz & Dr. K. Brandenburg GbR, Bonn, Germany) [29,30]. The mineralogical phases present were identified using Crystallography Open Database (COD) entries as a reference (Table 4).

NMR/MAS scans were recorded with a Bruker Avance-400 spectrometer (Bruker, Kontich, Germany). The recording conditions for the ^29^Si spectra were: resonance frequency, 79.5 MHz; rotational speed (NMR), 10 kHz; pulse length, 5 s; recycle delay, 10 s; external standard: tetramethylsilane (TMS). The respective conditions for ^27^Al were: resonance frequency, 104.3 MHz; rotational speed (NMR), 10 kHz; pulse length, 2 s; recycle delay, 5 s; external standard: (H_2_O)_6_^+3^.

Morphological and elemental microanalyses were conducted on an Inspect FEI Company scanning electron microscope (SEM) fitted with an energy dispersive X-ray (EDX) analyser and a Si/Li detector. The semiquantitative chemical analysis values shown are the mean of 10 analyses per spot.

FTIR spectra were recorded with a Bruker Alpha spectrometer operating with a He–Ne laser at a range of 450 cm^−1^ to 4000 cm^−1^. The spectroscopy-grade 1% KBr pressed wafers and reference calcined clay samples were stored overnight at 250 °C to remove any adsorbed water.

## 3. Results

### 3.1. XRD–Rietveld Analysis

The XRD–Rietveld findings for the cements analysed are provided in Table 5, Table 6 and Table 7. Anhydrous phases C_3_S, C_2_S, C_3_A, C_4_AFe, and CaCO_3_ were clearly identified in the original OPC paste, in addition to 8% amorphous matter. Hydrated crystalline phases including C_4_ACH_11_ (traces), ettringite (10%), and portlandite (23%) first appeared in the 2-day paste. As those percentages rose over time, the 90-day values for the hydrated phases were traces for C_4_ACH_11_, 17–23% for ettringite, and 20–29% for portlandite. The upward pattern was simultaneous with a slight increase in calcite and substantial growth in the amount of amorphous matter (Table 5).

The pastes bearing 1:2 and 2:1 HcG/glass binary blends at a replacement ratio of 7% (Table 6) exhibited the same mineralogical phases as observed in the reference OPC at all ages, although with lower anhydrous phase contents. That fact, along with the rise in the relative concentrations of ettringite, portlandite, and amorphous phase (C-S-H gels), confirmed accelerated hydration due to the filler effect induced by the additions, more prominently in the cement pastes with the smaller proportion of glass (2:1).

The effect of the presence of glass in the blend with the calcareous concrete waste (HcG) could be tracked by monitoring the variation in calcite and the amorphous phase, which declined in the 2:1 blend but rose in the blend with a higher glass content (1:2) [11,21].

From the outset, both (2:1 and 1:2) 7% HsT/glass blends exhibited the same mineralogical phases as the 7% HcG/glass blends, although the amounts forming at each age differed (Table 7). The presence of the binary siliceous blends lowered portlandite and ettringite formation and, beginning in the 28-day pastes, 5% to 7% carboaluminate was detected, whilst the amount of glass added had scarcely any effect. This type of waste was also observed to have a direct effect on the amorphous matter content, which, at 40%, was much higher than in any of the other pastes analysed. That effect denoted the pozzolanicity of siliceous concrete waste, which would enhance C-S-H gel formation within a few minutes of hydration.

### 3.2. SEM/EDX Morphological Analysis

SEM/EDX exploration identified non-crystalline phases in all the pastes associated with fibrous C-S-H gels which, once formed, served as a growth substrate for ettringite fibres and portlandite plates (Figure 1). All the phases at issue grew in size with time [28].

The 7% HsT/glass sample (Figure 2, left) was observed to be denser, more compact, and less porous than 7% HcG/glass paste (Figure 2, right), which was more porous, particularly between the portlandite plates and ettringite fibres, with the appearance of many interphase voids. The appearance of ettringite occurs throughout the sample and its fibrous nature is what produces the appearance of holes as the entire space is not covered by a steric impediment in growth. It must be taken into account that the observation has been made on a small scale and that it is a reproduction of what happens when the observation is larger.

EDX microanalysis of the C-S-H gels showed both blended pastes to contain aluminium. The C-S-H gels in the 2:1 blends exhibited lower Ca/Si and Al/Si ratios than found in OPC gels (Table 8), although the difference was narrower in the pastes’ calcareous concrete waste (7% HcG/glass). Those findings denote silicon uptake in the C-S-H gels, more notably in the pastes prepared with siliceous concrete waste.

### 3.3. TG/DTA Analysis

The variations in the TG/DTA thermogravimetric readings for the 28-day and 90-day ternary hydrated cements are depicted in Figure 3. All the pastes exhibited very similar thermal behaviour, with three distinct endothermal zones. A first was recorded at 50 °C to 300 °C, temperatures typically associated with the dehydroxylation of the main hydrated phases, formed during cement hydration, as well as with the pozzolanic reaction [29]. The most prominent band in this zone (112 °C) would be attributable to C-S-H gel dehydroxylation, whilst ettringite would dehydroxylate at <100 °C, the bound water in carboaluminate at 155 °C, and carboaluminate at 180 °C to 210 °C. A third small endothermal band at 370 °C to 375 °C was also identified with C-S-H gel dehydroxylation.

The second endothermal zone, stretching from 400 °C to 550 °C and peaking at 463 °C, was typical of portlandite dehydroxylation and the third, at 550 °C to 750 °C with a peak close to 720 °C, of carbonate decarbonation [31].

By the TG/DTA analysis, all the 28-day and the 90-day cement pastes were similar, irrespective of the binary blend added. At 28 days, only a minor difference was observed in the endothermal range 50 °C/400 °C, where the OPC paste had a greater mass loss than the ternary pastes, possibly as a result of the dilution effect and low (HsT and HcG) CDW activity [11]. In the 90-day samples, the TG/DTA curves were nearly identical with no qualitatively perceptible differences.

The mass loss recorded from the TG/DTA curves at temperature intervals 50/400 °C, 400/550 °C, and 550/750 °C is listed in Table 9, which also gives the relative portlandite content of each sample, given the 7% replacement of the reference OPC.

According to the data in Table 9, mass loss in the ternary cements was slightly lower than in the OPC at 50 °C to 400 °C (C-S-H, ettringite and Ca(OH)_2_ dehydroxylation), with a somewhat wider difference at 28 days. That observation was related to the filler effect and low pozzolanicity of the binary blends, for the Ca(OH)_2_ values (400 °C to 550 °C) were practically the same. The effect was corroborated by the relative (to OPC) portlandite content values, which were greater (positive values) than they should have been after replacing 7% OPC with the additions at hand. The relative portlandite content was higher in the ternary pastes where glass prevailed (1:2) in the binary blend than where concrete waste prevailed (2:1). That observation would be related to portlandite insolubility, intensified by the uptake from the medium of the sodium ions present in the recycled glass [22].

Weight loss was greater at 50 °C to 400 °C in the 90-day materials, in keeping with the formation of more C-S-H gel with hydration time. Whilst loss was greater in the 28-day OPC than in the 28-day blended cements, the 90-day values were similar, denoting gel formation during the pozzolanic reaction.

### 3.4. FTIR Analysis

The results on hydration as monitored with FTIR (Figure 4) corroborated the XRD findings, while adding no further information on amorphous phase formation. The Si-O tetrahedral group bending vibrations peaking at 452 cm^−1^ and the same group’s stretching vibrations at 900 cm^−1^ to 1000 cm^−1^ denoted the presence of silicates. The samples with the 2:1 replacement exhibited the highest proportion of hydrated phase, attested to by the relative intensity of the portlandite bands at approximately 3645 cm^−1^ and the bands at approximately 3443 cm^−1^ associated with the OH groups in C-S-H gels. The 1:2 sample blends exhibited more intense carbonate (primarily calcite)-induced bands at 1414 cm^−1^ and 874 cm^−1^ [31] and a less intense band associated with portlandite OH groups (3645 cm^−1^), perhaps indicating its partial carbonation.

In another vein, Gismera et al. [32] contended that monocarboaluminate was present, attributing the signal at 3540 cm^−1^ to its OH groups. The wide signal with a small peak observed in this study may have been indicative of such a presence.

### 3.5. NMR Analysis

The ^29^Si NMR spectra for the 90-day blended cements containing 7% HcG/glass or HsT/glass at ratios of 2:1 or 1:2 are reproduced in Figure 5. Signals denoting the presence of the initial (anhydrous) Q^0^ units as well as the Q^1^ (-71ppm), Q^2^(1Al) (−81.5 ppm), and Q^2^ (−85 ppm) units found in the C-S-H gel [33] were present in all. The signal at −81.5ppm denoting Al uptake by the C-S-H gel was more intense in the 7% HsT (1:2) than in the 7% HcG (1:2) paste.

Table 10 compares the Q^n^ content and mean chain length (MCL) values for the ternary pastes drawn from the deconvoluted ^29^Si NMR spectra with the data for OPC published in an earlier paper [27].

The lower intensity of the Q^0^ signals associated with the initial anhydrous phases in the ternary pastes than in OPC was a sign of higher reactivity in the former. All the blended pastes took Al up into their C-S-H gels, an effect more prominent in the materials prepared with (1:2) HcG/glass and (2:1) HsT/glass, which were also the gels with the longest chain lengths. Al uptake into C-S-H gel depends on the amount of dissolved Al, Ca, and Si, as well as on the pH of the porous solution [34,35]. The presence of alkaline ions such as Na may also favour Al uptake in the gel to balance the charge difference between Al and the Si it replaces [36], possibly associated with the higher Na content in the waste than in OPC. Al(IV), taken up into silicate tetrahedra bridging sites, prevails at low Ca/Si ratios, whereas high Ca/Si ratios (approximately above 1.2) result in larger fractions of Al(V) and Al(VI), species associated with the C-(A)-S-H structure [33].

The ^27^Al NMR spectra for the 7% blended cements reproduced in Figure 6 exhibit clearly distinguishable Al(VI) and Al(IV) signals. All the pastes generated two Al(VI) signals, one at 13.4ppm attributed to ettringite and the other at 8.7ppm to calcium monocarboaluminate hydrate [36].

In addition, the ^27^Al NMR spectrum for the paste bearing siliceous concrete waste (HsT) exhibited a signal at 56 ppm attributed to non-reactive tecto- and phyllo-silicates [16], whilst the signal at 68 ppm was associated with the Al(IV) found in silicate chain bridging sites [37].

Samples with limestone residue have higher Imonocar/Iett values, indicating that they contain a higher percentage of carbonate phases. In the case of samples with siliceous residue, however, the ratio is lower, indicating a higher ettringite content. In the study carried out on the OPC samples with a single residue (HcG or HsT), it was observed that if the residue was HcG, what was formed was aluminates or carboaluminates in addition to ettringite, but if the residue added was of a siliceous nature (HsT), only ettringite was formed. In these samples, in which glass is added in addition to the limestone or siliceous residue, what is observed is that the formation of aluminates or carboaluminates is favoured in both cases (Figure 7).

## 4. Discussion

The synergies in a blend of pozzolans present in (calcareous and siliceous) waste concrete fines with recycled CDW glass were analysed in depth in the study reported in this article.

Ettringite and portlandite were the hydrated phases identified in the reference OPC paste. Their content, along with the amounts of C-S-H gels, rose with reaction time.

Partially replacing OPC with 7% of the binary blend, either HcG/glass or HsT/glass at ratios of 2:1 or 1:2, had no qualitative effect on the mineralogical phases produced with cement hydration. Quantitative changes were nonetheless observed. In the pastes with the higher HcG/glass ratio (2:1), ettringite, portlandite, and amorphous phase formation was enhanced, whereas in the pastes containing siliceous concrete (HsT/glass) C-S-H gel formation was favoured. In both cases, higher glass contents (1:2) delayed hydration, with a concomitant decline in the relative amounts of reaction product.

Cement hydration yielded portlandite as the primary reaction product, although the type of CDW used conditioned the content of the mineral and its variation over time. Portlandite content was higher in the blends containing calcareous waste and glass than in OPC at all ages. Hydration was accelerated in the 28-day materials in particular due to the filler effect prompted by the waste [16]. At later ages (90 days), the presence of glass at whichever proportion (2:1 or 1:2) delayed hydration due to the decline in portlandite solubility as a result of the sodium in the medium [37,38], which neutralised the pozzolanic effect in the binary blends of CDW. The pastes with siliceous concrete were slightly more pozzolanic than the ones with calcareous waste at both ratios (Table 9).

Ettringite, in turn, is highly sensitive to the composition of the medium, depending primarily on the presence of sulfur, pH, and calcite content. This phase usually forms by non-uniform nucleation on C-S-H gel substrates in the presence of sulfur ions [39,40], whilst the presence of calcite in the medium, which prevents its conversion to monosulfoaluminate, favours stabilisation [41].

Here, due to the presence of calcite in the waste itself, a component absent from OPC, the ettringite precursor monocarboaluminate was observed to form in all the blended pastes. This is consistent with earlier research [42], according to which a higher content of this phase, ettringite, was found in pastes prepared with calcareous concrete waste.

The type of CDW used in the cement pastes modified C-S-H gel composition relative to the gel found in OPC paste. The presence of the waste favoured silicon uptake in the C-S-H gels, as attested to by the shift in the ^29^Si NMR signals, most prominently in the mixes containing HsT/glass where the Ca/Si and Al/Si ratios declined (Table 8). Al uptake into the C-S-H gel, observed in both the OPC and all the blended pastes analysed but especially in the material (2:1) HsT/glass, tended to lengthen the respective chains, which would explain the fibre-like structure visible in the SEM micrographs.

## 5. Conclusions

Adding calcareous or siliceous waste concrete/waste glass blends to cement pastes affects the proportion although not the nature of the mineralogical phases present in the hydration product. Pastes bearing calcareous concrete waste (HcG/glass ratio) enhance ettringite, portlandite, and amorphous phase formation, whereas the generation of C-S-H gels is favoured in pastes containing siliceous concrete (HsT/glass). In both cases, higher glass contents (1:2) delay hydration, with a concomitant decline in the relative amounts of reaction product. In the mixes bearing glass waste, the presence of calcite determines the formation of the ettringite precursor monocarboaluminate, which also acts as calcareous filler. The nature of the waste affects C-S-H gel composition, with a decline of approximately 36% to 44% in the Ca/Si and 50% to 76% in the Al/Si ratios.

Despite the scientific advances reported here, further research is needed to analyse the effect of such microstructural changes on ternary eco-cement mechanical strength and durability.

## Figures and Tables

**Figure 1 materials-15-04661-f001:**
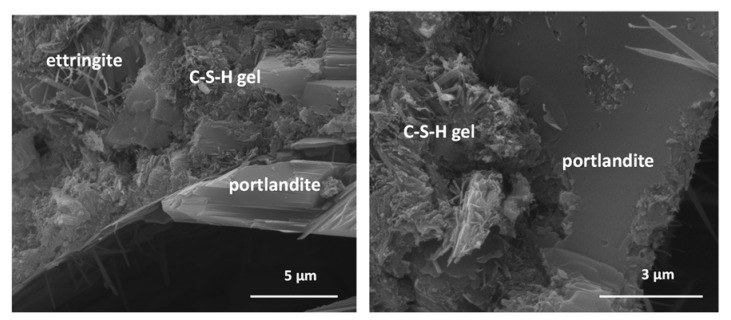
(**Left**) C-S-H gels, portlandite, and ettringite filling cracks. (**Right**) Detail of C-S-H gel fibres and portlandite formation on the gel.

**Figure 2 materials-15-04661-f002:**
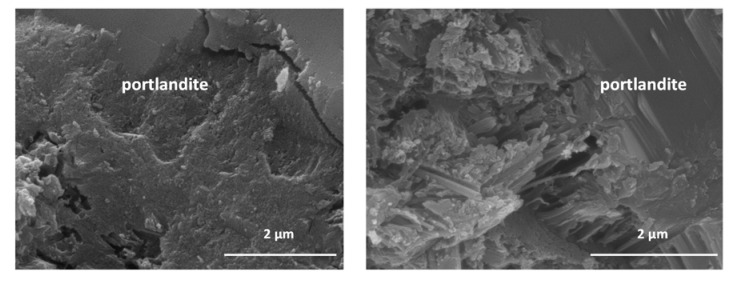
(**Left**) Detail of (2:1) blend compaction. (**Right**) HcG/glass cement.

**Figure 3 materials-15-04661-f003:**
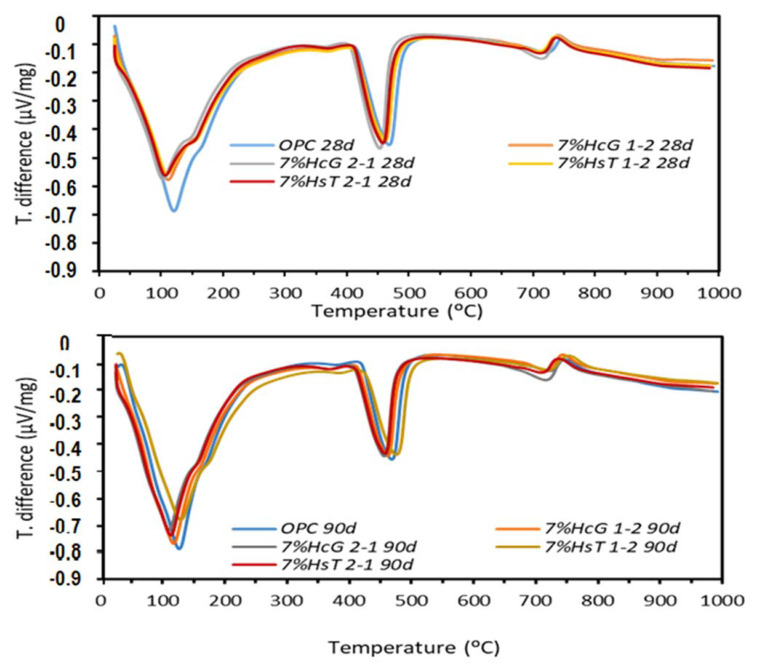
TG/DTA curves at 28 and 90 days of treatment from ternary hydrated cement.

**Figure 4 materials-15-04661-f004:**
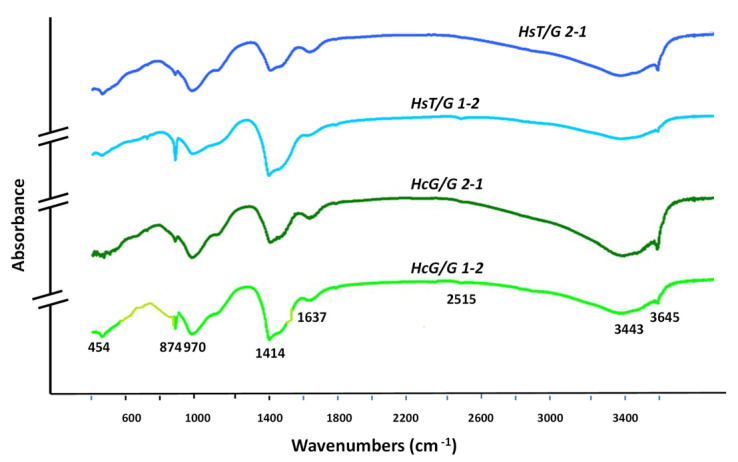
IR spectra for 90 days of blended cements.

**Figure 5 materials-15-04661-f005:**
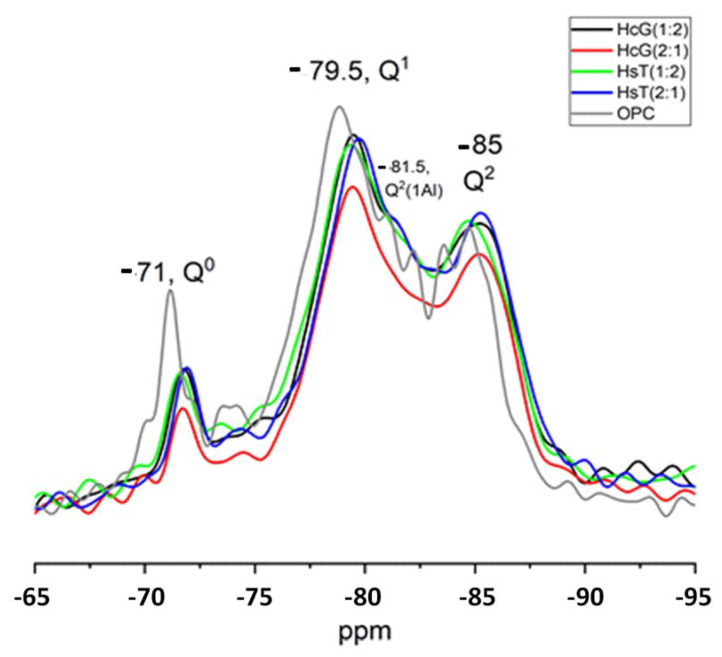
^29^Si NMR spectra for 90-day 7% HsT/glass and HcG/glass pastes.

**Figure 6 materials-15-04661-f006:**
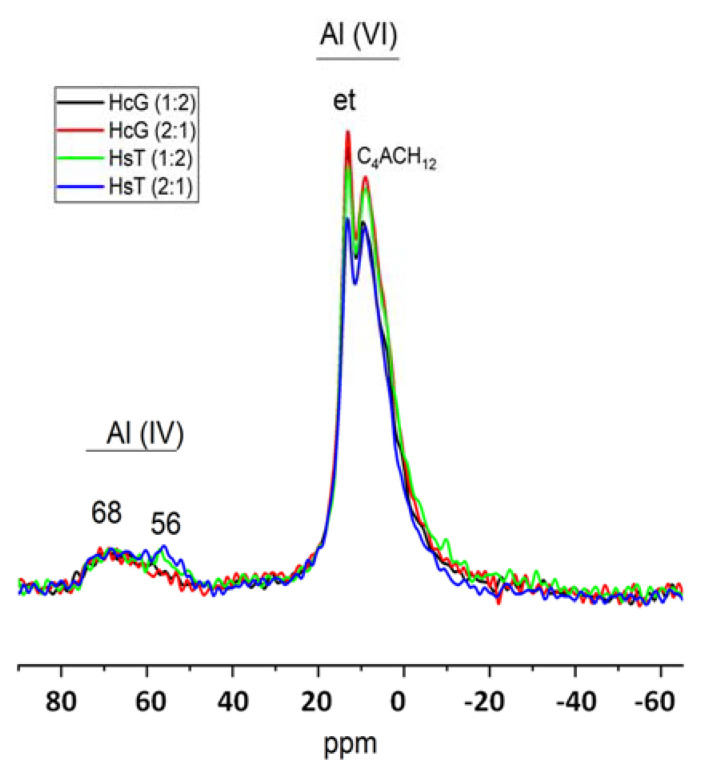
^27^Al NMR spectra for 90-day 7% HsT/glass pastes.

**Figure 7 materials-15-04661-f007:**
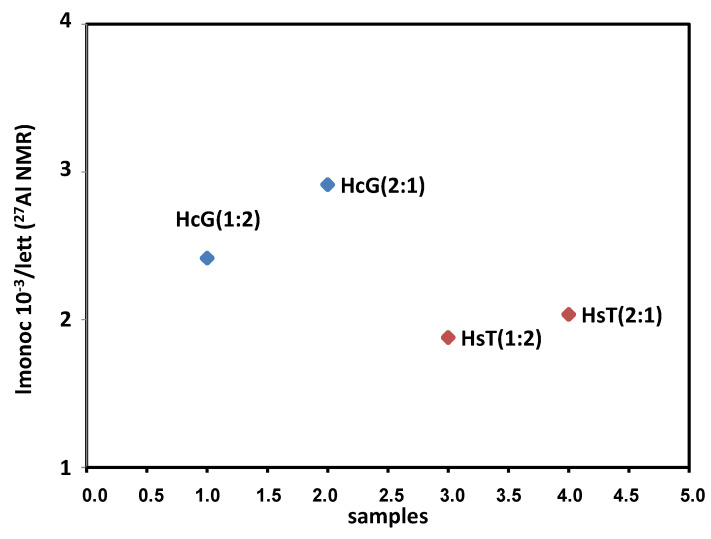
Ratio of the intensities of the ^27^Al NMR signals for monocarboaluminate and ettringite (8/13.4).

**Table 1 materials-15-04661-t001:** Cement nomenclature.

Cement	OPC (wt. %)	HsT (wt. %)	HcG (wt. %)	Glass (wt. %)
OPC	100	0	0	0
7% HsT/G 1:2	93	2.33	0	4.67
7% HsT/G 2:1	93	4.67	0	2.33
7% HcG/G 1:2	93	0	2.33	4.67
7% HcG/G 2:1	93	0	4.67	2.33

**Table 2 materials-15-04661-t002:** FRX-determined chemical composition (%) of the cements studied (LOI = loss on ignition).

(%)	OPC	7% HsT/G 1:2	7% HsT/G 2:1	7% HcG/G 1:2	7% HcG/G 2:1
SiO_2_	14.22	16.69	15.28	17.63	17.16
Al_2_O_3_	2.89	2.80	2.84	2.94	3.13
CaO	69.81	66.53	67.47	65.80	66.00
Fe_2_O_3_	3.70	3.51	3.51	3.54	3.57
MgO	0.93	1.06	1.00	1.06	1.01
SO_3_	3.36	3.16	3.17	3.19	3.25
Na_2_O	0.33	0.92	0.62	0.94	0.65
K_2_O	0.76	0.73	0.74	0.80	0.87
P_2_O_5_	0.14	0.13	0.13	0.13	0.14
TiO_2_	0.20	0.19	0.19	0.20	0.20
LOI	3.22	3.78	4.54	3.28	3.54

**Table 3 materials-15-04661-t003:** D10, D50, and D90 values (µm) for the cements studied.

	OPC	7% HsT/G 1:2	7% HsT/G 2:1	7% HcG/G 1:2	7% HcG/G 2:1
D10	2.07	2.04	1.94	1.97	1.95
D50	11.4	11.4	10.9	11.1	10.9
D90	34.8	34.8	33.9	34.6	34.2

**Table 4 materials-15-04661-t004:** Crystallography Open Database (COD) files used for Rietveld quantification.

Mineral	File Number
Alite	992,000,105
Belite	99,200,096
Calcite	992,000,080
Al_2_Ca_3_O_6_	992,000,100
Portlandite	992,000,046
Ettringite	992,000,085
C_4_ACH_11_	992,000,037

**Table 5 materials-15-04661-t005:** Mineralogical composition of OPC paste (XRD–Rietveld).

Phase (%)	Initial	2 Days	28 Days	90 Days
Alite	52	28	14	traces
Belite	20	15	10	5
C_3_A	9	5	traces	traces
C_4_AF	6	5	traces	traces
Calcite	5	7	14	11
Ettringite	n.d.	5	8	10
Portlandite	traces	17	26	28
C_4_ACH_11_	traces	traces	5	8
A. matter	8	18	23	38
Total	100	100	100	100
R_B_	12.3	14.5	10.6	9.7
χ^2^	7.9	8.3	7.6	7.5

n.d. = not detected; A. matter = amorphous matter; R_B_ and χ^2^, agreement factors.

**Table 6 materials-15-04661-t006:** Mineralogical composition of 7 % HcG/glass paste.

Phase (%)	(1:2) 2d	(2:1) 2d	(1:2) 28d	(2:1) 28d	(1:2) 90d	(2:1) 90d
Alite	6	7	n.d.	n.d.	n.d.	n.d.
Belite	5	5	n.d.	n.d.	n.d.	n.d.
C_3_A	traces	traces	n.d.	n.d.	n.d.	n.d.
C_4_AF	n.d.	n.d.	n.d.	n.d.	n.d.	n.d.
Calcite	25	23	26	16	25	14
Portlandite	23	34	27	38	30	39
Ettringite	10	11	15	23	17	23
C_4_ACH_11_	traces	traces	traces	traces	traces	traces
A. matter	31	20	32	23	28	24
Total	100	100	100	100	100	100
R_B_	16.8	12.2	15.8	13.7	19.8	16.3
χ^2^	5.7	7.1	7.9	6.8	8.3	8.5

n.d. = not detected; A. matter = amorphous matter; R_B_ and χ^2^, agreement factors.

**Table 7 materials-15-04661-t007:** Mineralogical composition of 7% Hst/glass paste.

Phases (%)	(1:2) 2d	(2:1) 2d	(1:2) 28d	(2:1) 28d	(1:2) 90d	(2:1) 90d
Alite	7	5	n.d.	n.d.	n.d.	n.d.
Belite	5	traces	n.d.	n.d.	n.d.	n.d.
C_3_A	traces	traces	n.d.	n.d.	n.d.	n.d.
C_4_AF	n.d.	n.d.	n.d.	n.d.	n.d.	n.d.
Calcite	38	38	39	36	30	33
Portlandite	9	11	15	17	19	24
Ettringite	traces	5	5	7	5	10
C_4_ACH_11_	n.d.	n.d.	n.d.	traces	5	7
A. matter	41	41	41	40	41	26
Total	100	100	100	100	100	100
R_B_	19.6	15.4	17.8	19.3	18.9	21.3
χ^2^	7.7	8..7	6.4	8.6	6.8	9.7

n.d. = not detected; A. matter = amorphous matter; R_B_ and χ^2^, agreement factors.

**Table 8 materials-15-04661-t008:** 90-day OPC and C-S-H gel ratios.

	OPC	7% HcG/Glass (2:1)	7% HsT/Glass (2:1)
Al_2_O_3_/SiO_2_	0.46	0.23	0.11
CaO/SiO_2_	3.68	2.37	2.06

**Table 9 materials-15-04661-t009:** Mass loss (%) by temperature range and relative Ca(OH)_2_ content (%) in the pastes studied.

%	50/400 °C (H_2_O)	400/550 °C (H_2_O)	550/750 °C (CO_2_)	Rel. Ca(OH)_2_
OPC 28d	16.25	5.05	2.66	0
OPC 90d	18.40	5.09	2.34	0
7% HcG 1-2 28d	14.50	5.01	2.70	+6.7
7% HcG 1-2 90d	18.65	5.03	2.75	+6.3
7% HcG 2-1 28d	14.31	5.06	3.24	+5.7
7% HcG 2-1 90d	17.90	4.94	3.13	+4.4
7% HsT 1-2 28d	14.23	4.91	2.71	+4.6
7% HsT 1-2 90d	17.21	5.05	2.17	+6.6
7% HsT 2-1 28d	14.14	4.86	2.93	+3.5
7% HsT 2-1 90d	18.05	4.92	2.47	+4.0

**Table 10 materials-15-04661-t010:** 90-day Q^n^ signal deconvolution and MCL compared with OPC.

	OPC	HcG (1:2)	HcG (2:1)	HsT (1:2)	HsT (2:1)
Q^0^	15.20	8.20	4.40	5.10	5.20
Q^1^	47.60	32.30	39.30	39.00	37.10
Q^2^ (1Al)	1.70	21.10	17.20	19.00	22.90
Q^2^	35.50	38.40	39.10	36.90	34.90
MCL	3.56	6.34	5.31	5.35	5.73

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
