# Peer review of "Concrete/Glass Construction and Demolition Waste (CDW) Synergies in Ternary Eco-Cement-Paste Mineralogy"

_materials, 2022, doi:10.3390/ma15134661_

Round 1

Reviewer 1 Report

The introduction section gives many paragraphs. I suggest the authors to combine these paragraphs and classify the literature according to their contents.

Author Response

We appreciate your comments that enrich the work and we proceed to answer the questions presented. Comments are highlighted in the text

The introduction section gives many paragraphs. I suggest the authors to combine these paragraphs and classify the literature according to their contents.

The introduction has been reorganized.

Reviewer 2 Report

a)      The title was wrong – ‘construction’

b)     I suggest new references (replacement 3-7) about classic SCM (supplementary cementitious materials), for example – FA/MK/SF (fly ash, metakaolin and silica fume).

c)      The lines 49-51 – shown just one case about rule or regulation on structural legislation. I suggest one case from EU and another USA or other countries.

d)     The words “given” and “bearing” are recured 4 and 18 times, respectively.

e)      Line 62, change ask for ash.

f)       Line 64, I suggest a review of the information, because the text don´t agree with de reference.

g)     Line 78 and 82 – adjust of the letter.

h)     Line 108 – review of specimens of prismatic value.

i)       Line 194 – the statement “blends exhibited lower Ca/Si and Al/Si “in Table 8, are showing in weigh of oxides or chemical elements?

j)        Table 9: I suggest that authors make a statistical analysis through standard deviation upon the data to explain the results, the values ​​are very closer.

k)     FTIR – The affirmation in the line 246 and 247 don´t agree with reference description number 31. This article don´t have FTIR analyses.

l)       Figure 4 - The wavenumber axis don´t agree with the graphics results, for example the axis shows wavenumber 3400 and the results 3645, in the same region in the graphic.

m)   Figure 5 – I suggest that OCP results will be putting together to promote the effective comparison.

n)     Figure 6: What kind of ratio of Ca/Si ration (weight or molar). In the article 36, the results are based on molar ratio. Do you can make a real comparison between the results.

o)     Figure 7: I suggest improving the figure, I can´t to understand what means the results.

p)     After this review, you can improve the discussion and conclusion.

I recommended these comments as contribution in your paper.

Congratulations for your research.

Best regards.

Author Response

We appreciate your comments that enrich the work and we proceed to answer the questions presented. Comments are highlighted in the text

The title was wrong – ‘construction’

Thank you very much it has been corrected

I suggest new references (replacement 3-7) about classic SCM (supplementary cementitious materials), for example – FA/MK/SF (fly ash, metakaolin and silica fume).

New references have been introduced in the revised version according to your suggestions.

Castro M.I, de Silva J.H. Effect of nanosilica/MK ratio on the calcium aluminate silicate hydrate (CASH) formed in ternary cement pastes. J. Build. Engin. 38(2021)102220)

Xu G., Shi, X. Characteristics and applications of fly ash a sustainable construction material: A state of art review. Resourc. Conserv. Recy. 136(2018)95-109

Snelson, D.G., Wild, S., O’Farrell, M. Heat of hydration of Portland cement-MK-FA blended. Cem. Concr. res. 38 (2008)832-840

Abdelmelek, N., Lubloy, E. Flexural strength of silica, fly ash and metakaolin of hardened cement paste after exposure to elevated temperatures. J. Therm. Analy. Calor. 14(2022)7159-7169

The lines 49-51 – shown just one case about rule or regulation on structural legislation. I suggest one case from EU and another USA or other countries.

Three international normative references have been included in the text.

ACI 130R-19: Report on the role of materials in sustainable concrete construction, American Concrete Institute; Farmington Hills, MI, USA, 2019

Standard DIN 4226-101. Recycled Aggregates for Concrete in Accordance with DIN EN 12620-Part 101: Types and regulated dangerous substances, German Institute for Standardization, Belin, Germany, 2017

Standard BS 8500-2. Concrete-Complementary British Standard to BS EN 206. Specification for constituent materials and concrete (+A2:2019), British Standards Institution, London, UK, 2015

 The words “given” and “bearing” are recured 4 and 18 times, respectively.

It has been corrected in the text

Line 62, change ask for ash.

word corrected

Line 64, I suggest a review of the information, because the text don´t agree with de reference.

The references have indeed been reviewed and updated with the subject matter.

Aprianti, E., Shafigh, P. Bahri, S. Nodek, J. Supplementary cementitious materials origin from agricultural wastes- A review. Constr. Build. Mater. 74(2015)176-187.

Liu, G., Florea, M.V.A., Brouwers, H.J.H. The hydration and microstructure characteristics of cement pastes with high volume organic contaminated waste glass powder. Constr. Build. Mater. 187(2018)1177-1189

Sanchez de Rojas, M.I., Frias, M. The pozzolanic activity of different Materials, its influence on the hydration heat in mortars. Cem. Concr. Res. 26(1996)203-213.

Line 78 and 82 – adjust of the letter.

The letter has been adjusted

Line 108 – review of specimens of prismatic value.

Indeed, by mistake, m3 appears instead of cm3. Modified in the text

Line 194 – the statement “blends exhibited lower Ca/Si and Al/Si “in Table 8, are showing in weigh of oxides or chemical elements?

In weigh of oxides

Table 9: I suggest that authors make a statistical analysis through standard deviation upon the data to explain the results, the values are very closer.

We do not understand the question exactly. Table 9 shows contents obtained from TG losses of point samples and although the values are similar, they cannot be analyzed with standard deviations because they

FTIR – The affirmation in the line 246 and 247 don´t agree with reference description number 31. This article don´t have FTIR analyses.

The reference has been changed to

Lin, W.; Zhou, F.; Luo, W.; You, L. Recycling the waste dolomite powder with excellent consolidation properties: Sample synthesis, mechanical evaluation, and consolidation mechanism analysis. Constr. Build. Mater. 2021, 290,123198.

Figure 4 - The wavenumber axis don´t agree with the graphics results, for example the axis shows wavenumber 3400 and the results 3645, in the same region in the graphic.

Thank you very much. The figure is new and collects the specific information

Figure 5 – I suggest that OCP results will be putting together to promote the effective comparison.

The figure is new

Figure 6: What kind of ratio of Ca/Si ration (weight or molar). In the article 36, the results are based on molar ratio. Do you can make a real comparison between the results.

I think you are referring to Figure 7, not 6. In Figure 7 are data of intensities determined by 27Al NMR deconvolution, they would not be comparable with the data of paper 36 that come from the analysis of the concentrations in the solutions.

 Figure 7: I suggest improving the figure, I can´t to understand what means the results.

The figure has been changed and the paragraph rewritten.

Reviewer 3 Report

I must say this is very interesting topic which is described in the manuscript. Glass and demolition wastes donstruction into concrete material which creates ternary eco-cement paste is novel and important for the future of the entire construction industry. New approach how to improve some of the properties and use of waste are always welcome. The application sustainable technologies in these new "eco- products" are currently important as well from the perspective of circular economy criteria.

This manuscript describes an interesting way how to use glass and demolition waste in concrete and produce new eco-cement/concrete product. I have just a few recommendations and questions for authors that need to be discussed.

1) There is a few typographical errors and missing symbols for example at 3.3. TG/DTA analysis article is missing ° before C few times. Or at line 138 is incorrectly used.

2) For the described XRD Rietveld analysis I think it would be interesting and more correct if the accuracy of the fitting for the evaluation of the samples were given.

3) Results from SEM/EDX morphological analysis, is well discussed, and described however I recommend giving more details, for example how homogeneous the structure is or is not at lower magnification and whether the distribution of ettringite formation is uniform or whether it only appears in certain sections.

I would also like to know if the less porous structure in 7% Hcg/Glass has any effect on the mechanism of mineral growth on hydration.

4) figure 7 and the text below above is slightly inconsistent with the rest of the article, the graph looks very different, and it might be a good idea to link it to the table with the measured results (personally I had a problem with understanding the labels).

5) Discussion, I little miss this topic in the discussion, but have you thought about using a different cement matrix with a lower content of C3S and C3A minerals, which together with sulphate ions increase the chance of Ettringite formation for future research?

6) As not native English speaker, I don't feel to be fully qualified to judge English language and style. But the text has been fully readable to me and technically easy to understand.

7) Large amount of the reference is from Spanish speaking region. It is strange the research in this field is not performed in such quantity in the rest of the world.

Author Response

We appreciate your comments that enrich the work and we proceed to answer the questions presented. Comments are highlighted in the text

There is a few typographical errors and missing symbols for example at 3.3. TG/DTA analysis article is missing before C few times. Or at line 138 is incorrectly used.

Thank you very much for the appreciation. It has been fixed.

For the described XRD Rietveld analysis I think it would be interesting and more correct if the accuracy of the fitting for the evaluation of the samples were given.

Tables 5, 6 and 7 give the adjustment factors of the Rietveld method, which are RB and X2. With them you can appreciate the goodness of the analysis, which as you can see is quite good.

3) Results from SEM/EDX morphological analysis, is well discussed, and described however I recommend giving more details, for example how homogeneous the structure is or is not at lower magnification and whether the distribution of ettringite formation is uniform or whether it only appears in certain sections.

I would also like to know if the less porous structure in 7% Hcg/Glass has any effect on the mechanism of mineral growth on hydration.

The following paragraph has been added to better explain the suggested issues.

“The appearance of ettringite occurs throughout the sample and its fibrous nature is what produces the appearance of holes as the entire space is not covered by a steric impediment in growth. It must be taken into account that the observation has been made on a small scale and that it is a reproduction of what happens when the observation is larger”.

Figure 7 and the text below above is slightly inconsistent with the rest of the article, the graph looks very different, and it might be a good idea to link it to the table with the measured results (personally I had a problem with understanding the labels).

Added timely reviewer clarification: “and it is related to the values shown in table 10 with the signals at 90 days and their deconvolution”.

Discussion, I little miss this topic in the discussion, but have you thought about using a different cement matrix with a lower content of C3S and C3A minerals, which together with sulphate ions increase the chance of Ettringite formation for future research?

Further investigation can be done with the suggestions. However, and taking into account that the materials are the CDW residues that are available, the research is forced to study some materials that are not chosen. On later occasions, materials with lower content of C3S and C3A can be selected.

As not native English speaker, I don't feel to be fully qualified to judge English language and style. But the text has been fully readable to me and technically easy to understand.

Thank you very much; it comes from the translation of a professional.

Large amount of the reference is from Spanish speaking region. It is strange the research in this field is not performed in such quantity in the rest of the world.

The research group is a pioneer in this research. At present, specific investigations are appearing that reproduce those of the group in other nationalities.